# Uncertainties, work conditions and testing biases: Potential pathways to misdiagnosis in point-of-care rapid HIV testing in Zimbabwe

**Morten Skovdal**[1]\*, **Nadine Beckmann**[2], **Rufurwokuda Maswera**[3],
**Constance Nyamukapa**[3,4], **Simon Gregson**[3,4]

**1** Department of Public Health, University of Copenhagen, Copenhagen, Denmark, **2** Centre for Research in Evolutionary, Social and Inter-Disciplinary Anthropology, University of Roehampton, London, United Kingdom, **3** Manicaland Centre for Public Health Research, Biomedical Research and Training Institute, Harare, Zimbabwe, **4** Department of Infectious Disease Epidemiology, Imperial College London, London, United Kingdom

\* m.skovdal@gmail.com, m.skovdal@sund.ku.dk

**Data Availability Statement:** Data cannot be shared publicly because of ethical restrictions. These restrictions are in place because the

## Abstract

Disconcerting levels of misdiagnosis are common in point-of-care rapid HIV testing programmes in sub-Saharan Africa. To investigate potential pathways to misdiagnosis, we interviewed 28 HIV testers in Zimbabwe and conducted weeklong observations at four testing facilities. Approaching adherence to national HIV testing algorithms as a social and scripted practice, dependent on the integration of certain competences, materials and meanings, our thematic analysis revealed three underlying causes of misdiagnosis: One, a lack of confidence in using certain test-kits, coupled with changes in testing algorithms and inadequate training, fed uncertainties with some testing practices. Two, difficult work conditions, including high workloads and resource-depleted facilities, compounded these uncertainties, and meant testers got distracted or resorted to testing short-cuts. Three, power struggles between HIV testers, and specific client-tester encounters created social interactions that challenged the testing process. We conclude that these contexts contribute to deviances from official and recommended testing procedures, as well as testing and interpretation biases, which may explain cases of misdiagnoses. We caution against user-error explanations to misdiagnosis in the absence of a broader recognition of how broader structural determinants affect HIV testing practices.

## Introduction

Extraordinary gains have been made in the scale-up of HIV testing and treatment in sub-Saharan Africa, contributing to significant reductions in AIDS-related deaths [1]. A global campaign to ensure that 90% of all people living with HIV know their status has dramatically increased the number of people being tested. However, there is growing evidence that large numbers of people in sub-Saharan Africa may receive incorrect information about their HIV status due to diagnostic or misclassification errors in HIV rapid testing programs [2–5]. A

qualitative data contain potentially identifying or sensitive information, and because we do not have consent from the participants to share their transcripts with the wider public. Under these conditions, the Medical Research Council of Zimbabwe have imposed restrictions on the public sharing of data. Data is however available from the Biomedical Research and Training Institute (contact via http://www.manicalandhivproject.org/manicaland-data) for researchers who meet the criteria for access to confidential data.

**Funding:** Research reported in this publication was supported by the Bill and Melinda Gates Foundation under award number OPP1131208. C. N. and S.G. acknowledge joint MRC Centre for Global Infectious Disease Analysis funding from the UK Medical Research Council and Department for International Development (MR/R015600/1). The funders have played no role in study design, data collection and analysis, decision to publish, or preparation of the manuscript URL1: https://www.gatesfoundation.org/ URL2: https://mrc.ukri.org/funding/science-areas/global-health-and-international-partnerships/funding-partnerships/mrc-dfid-concordat/.

**Competing interests:** The authors have declared that no competing interests exist.

systematic review of 64 studies investigating misdiagnosis in rapid diagnostic HIV testing concluded that an average of 0.4% and 3.1% of adults being tested receive a false negative or false positive test result respectively [6]. While there is uncertainty associated with all medical testing and diagnosis, and misdiagnosis *is* a relatively rare occurrence, the sheer (and growing) number of people getting tested every year adds up to a significant absolute number of people being misdiagnosed. According to Johnson et al. [7], an estimated 93,000 people could receive an incorrect HIV diagnosis every year. False-negative clients may experience a potentially life threatening delay in accessing treatment, whilst false-positive clients, in an era of test-and-treat, may be put onto lifelong treatment needlessly, and risk facing the often devastating consequences of HIV-related stigma, potential side-effects and psychological distress [8]. Misdiagnoses can thus have dire negative consequences for people's health and survival, their social life and onward transmission of HIV infection.

To prevent misdiagnoses, much work has gone into evaluating the procedures for performing and interpreting HIV rapid tests, as well as how best to document, report and deliver test results. Based on such evidence, the WHO has released guidelines and recommendations for improving the quality of HIV-related point-of-care testing, ensuring the reliability and accuracy of test results [9, 10]. Nonetheless, the recent systematic review of 64 studies reporting on misdiagnosis of HIV found 37 studies to report on HIV testing programmes that applied testing algorithms that differed from WHO recommendations, and in 25 studies HIV testers were found to perform and interpret test results erroneously [6]. The review also noted significant issues with supply management and stock-out, as well as use of expired or inappropriate testing kits [6]. Such findings demonstrate that sources of misdiagnosis are multiple, spanning provider, facility and system-level factors. Although user error (e.g. documentation and reporting, performing the test, interpreting results, application of testing algorithm, management) features as a common source of misdiagnosis, we know strikingly little about how clinicians and primary counsellors who perform rapid HIV testing experience, perceive and explain misdiagnosis. Whilst much has been written about the views and experiences of clinical staff in the provision and management of provider-initiated HIV testing [11], including the stress experienced by testers [12, 13], this is, to our understanding, the first study that sets out to draw on their accounts and practices to understand potential pathways to misdiagnosis.

The study takes place in Zimbabwe, where point-of-care rapid HIV tests have played a central role in the country's extraordinary progress towards the goal of ensuring that 90 percent of people living with HIV know their status. In 2018, 94% of women and 86% of men living with HIV in Zimbabwe knew their HIV status [14]. Such progress has been achieved through a concerted and accelerated effort by the Ministry of Health in Zimbabwe to avail free HIV testing and counselling. A recent review of progress pertaining to the implementation of national HIV policies in Zimbabwe note that the mean number of health-facility HIV testing visits rose from 202 in 2013 to 489 in 2015 –despite reductions in numbers of healthcare workers [15]. The review notes that this progress has been achieved through a mix of task-shifting, a decentralisation of HIV testing services, including community-based HIV testing services, and an expansion of prevention of mother-to-child transmission (PMTCT) programmes, all of which has been made possible through the availability of rapid HIV tests. This rapid scale-up has, however, not been without its problems, with most health facilities reporting at least one test-kit stock-out in the prior year [15]. It is within this context we set out to explore potential pathways to misdiagnosis.

## Rapid diagnostic HIV testing: A social and scripted practice

This study takes inspiration from science and technology studies to investigate potential pathways to misdiagnosis. We approach rapid diagnostic HIV testing as a social and scripted

practice involving both human and non-human actors. As HIV only becomes visible through testing (with the exception of a few late presenters who may exhibit a number of HIV-related symptoms), there is no way for HIV testers to know that they have misdiagnosed when they record the results of HIV testing. Their experiences, perceptions and explanations of misdiagnoses will therefore naturally centre on whether and how their testing practices follow or deviate from the 'script' of rapid diagnostic HIV testing. According to Akrich [16], all technical objects have inscribed a script which details and guides the practice of using the object. The script in this case refers to the Zimbabwe national HIV testing strategy (see Fig 1) and each of the standard operating procedures (SOPs) that accompany rapid diagnostic HIV tests, devised

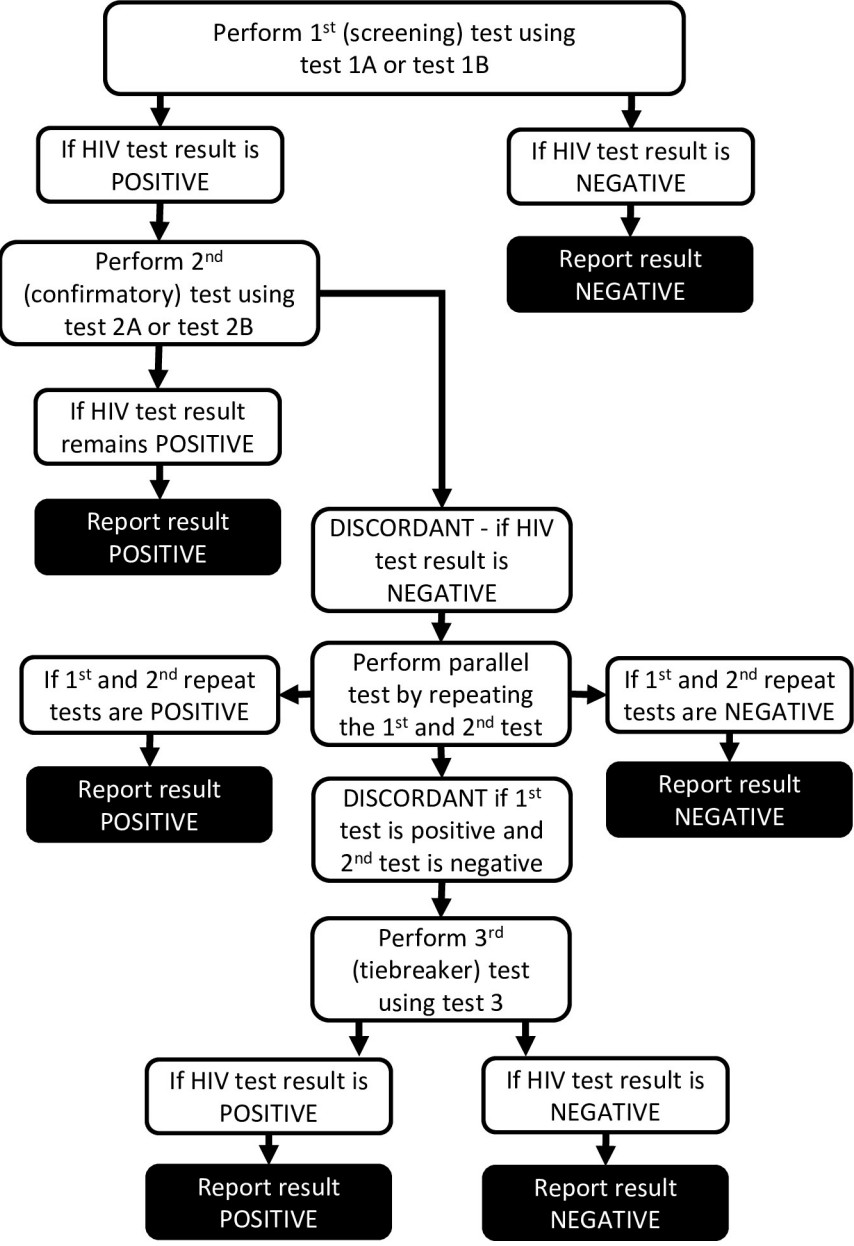

**Fig 1. The Zimbabwe national HIV testing strategy from 2018–2019, with the brand-names of the rapid diagnostic tests anonymised.**

to circumvent misdiagnosis. The national testing strategy illustrated in Fig 1 was introduced in 2018, shortly before the start of this study. It is depicted as a strategy as we have anonymised the brand-names that populate the national testing algorithm. The testing strategy stipulates that if the first test is positive, a different and second test needs to be run. At the time of the study, test 1A and test 2A were the most common tests used for first and second testing respectively. If the first and second tests produce a discordant result, the two tests are repeated in what is called parallel testing. If the test results remain discordant, the strategy recommends a tie-breaker test, using test 3. Parallel testing and the tie-breaker test are new to the 2018 testing strategy, which replaced an algorithm introduced in 2013. A new strategy reflecting WHO recommendations was introduced in Zimbabwe in 2019, removing the tie-breaker test, instead referring the client for re-testing after two weeks. This regular change in testing algorithms highlights how dynamic and responsive they are to the latest science and new HIV tests (their efficacy, pricing etc.) and provides critical context for understanding pathways to misdiagnosis.

The script, and its dynamic nature, determine the relationship between testers and clients, the spaces in which the rapid HIV tests are stored and administered, the temporality of each test, testing practices, as well as the values surrounding the mode of doing 'good HIV testing' [17]. Much work has gone into developing these scripts, often by people who are far removed from the contexts in which they are applied. So while the people and organisations who fund, engineer, or study rapid HIV tests make qualified assumptions and projections about who the end users are, and inscribe these into their engineering of the rapid HIV tests, SOPs, and testing algorithms, they are unlikely to capture the reality of each and every context [16]. Nonetheless, the fact that rapid HIV testing is scripted encourages us to examine how clinic-staff (dis) engage with the HIV testing algorithm, and assess its 'goodness of fit' to settings in Zimbabwe. We approach their engagement with the script, and the bundle of practices associated with rapid diagnostic HIV testing, as *social practices* [18, 19]. We thus move beyond narrow person- and user-focused paradigms of rapid HIV testing, and instead ask questions about how the integration (or otherwise) of core elements, such as competences, materiality, and meanings [18], determine the capacity of HIV testers to engage with recommended HIV testing practices (the script).

## Materials and methods

This qualitative study explored potential pathways to misdiagnosis from the perspective of clinicians and primary counsellors in Zimbabwe who performed rapid HIV testing on a daily basis. The study formed part of a larger mixed-methods study ('the misclassification study') that examined the sources and consequences of misclassification errors in rapid HIV test algorithms. The misclassification study was implemented by the Manicaland Centre for Public Health Research [20], and approvals for the study were granted by the ethical review boards of the Medical Research Council of Zimbabwe (MRCZ/A/1865) and Imperial College London (15IC2797). We obtained informed and written consent from all study participants and have used pseudonyms throughout to ensure their anonymity.

### Study context and participants

This misclassification study was embedded within a national HIV surveillance survey involving 62 antenatal clinics across Zimbabwe, purposefully selected to be reasonably representative of the country. The study provided new data on levels of misdiagnosis, and it was from this data we have sampled 14 health facilities, spread across six districts, namely Chipinge, Gutu, Buhera, Harare, Mudzi and Rushinga. The health facilities covered a mix of district hospitals

and local clinics that performed either well above or below national average rates of misdiagnosis. Logistics and geographical proximity were also considered when purposefully sampling the health facilities. Two or three rapid HIV testers were recruited from each facility. The inclusion criteria was that they had to be designated rapid HIV testers at their facility, and perform rapid HIV tests on a regular basis. As more than two or three HIV testers were eligible for participation at many of the health facilities, we invited the local Matron or Sister in charge to select the participants based on their experience and availability to participate in an interview on the day of our visit. Twenty-eight clinic-staff agreed to participate in interviews (see characteristics in Table 1). No participants refused to participate or subsequently withdrew. We also carried out week-long participant observations in four of the health facilities. Observations were being carried out from Monday through to Saturday from 9am to 4pm. This amounted to 24 observational visits.

## Data collection and analysis

Four experienced qualitative researchers, one male and three females, conducted the interviews primarily in Shona, the mother tongue language of all four researchers. A few interviews were conducted in English due to participant preferences. The researchers all had higher education degrees in social work. The interviews were semi-structured, guided by a topic guide that sought to elicit information about how and when rapid tests were introduced in their clinic, how they store and manage test-kits, and their experiences of administering the test-kits. The topic guides also asked about the potential causes and consequences of misdiagnosis. The interviews were conducted at the health facilities and took an average of 60 minutes. All participants received a t-shirt to compensate them for their time in assisting with the study. The interviews were recorded digitally and transcribed into English by the local research team. This involved translating the Shona interviews into English during the transcription process. To ensure nothing was lost in the transcription process, a quarter of the interview recordings were re-listened to by another researcher whilst reading the transcripts. No issues were noted from these quality checks. The quotations presented in this paper have been further edited to aid readability.

The participant observations were carried out by five researchers. The observations were steered by an observation guide focusing on the HIV testing practice, with attention paid to actors, objects, communication, affordances, and the spatial and temporal context of rapid

**Table 1. Participant characteristics.**

| Participant characteristics | | N |
|---|---|---|
| *Health facility performance* | | |
| | Low performing facility | 22 |
| | High performing facility | 6 |
| *Gender* | | |
| | Female | 20 |
| | Male | 8 |
| *Occupation* | | |
| | HIV counsellors | 17 |
| | Nurses | 10 |
| | Lab technician | 1 |
| *Healthcare setting* | | |
| | Hospitals | 8 |
| | Clinics | 6 |

HIV testing. All observations began with a tour of the facilities, with a particular emphasis on the testing rooms, the pharmacy, and the drug storage rooms. This was followed by a week of observing HIV testing practices in different departments. Care was taken to observe a number of different testers in each health facility, including nurses and primary counsellors who had been trained in applying rapid HIV tests. Data from the observations was captured in the form of field diaries and compiled into observation reports for each of the four observation sites. Data were generated in May 2018, and stored in secure password-protected locations.

Interview transcripts and observation reports were imported into NVivo 12 for thematic coding and interrogation [21]. Inductive and in-depth coding of 10 randomly selected transcripts was done by MS and NB independently from one another. The two coding frameworks were compared to each other, and a single coding framework was devised and applied to the remaining data. Only a few iterations were subsequently made to the coding framework. This process led to the development of 159 codes, or basic themes. The basic themes were subjected to a thematic network analysis [21, 22], and clustered together into 22 organising themes (or parent nodes in NVivo). The organising themes were grouped together into 4 global themes (grandparent nodes in NVivo). Ongoing data analysis, emerging hypotheses and our conceptual framework (detailed above) were discussed amongst the authors. This led to a further layer of thematic organising of basic themes, focusing on potential sources of misdiagnosis. In this paper we report on nine basic themes, selected because they detail the social and scripted nature of rapid diagnostic HIV testing practices, and offer insight into how the distribution of uncertainties (lack of competence), difficult work conditions (material and resource context), and social interactions (relational meanings) impact HIV testing practices and adherence to the national testing algorithm. We clustered the basic themes into these three organising themes (see Table 2), which form the structure of our presentation of the results.

## Results

According to our interviewees, misdiagnosis was a rarity, and nearly half of them stated they had never heard of such an occurrence, highlighting the invisibility of misdiagnoses. Overall, our interviewees felt that rapid tests were user-friendly and easy to use. The HIV testers we observed appeared overall very confident in handling the tests. Nonetheless, despite this apparent confidence, our interviewees also expressed uncertainty about their use of the tests, and observations captured some deviations from SOPs and the national testing algorithm. We also noted the considerable constraints high workloads posited for keeping up good quality in testing, all of which may affect their adherence to recommended testing practices.

**Table 2. Thematic network of emerging findings.**

| Basic themes (sources of misdiagnosis) | Organising themes |
|---|---|
| 1. Some test-kits cannot be trusted and produce faint or unclear lines | A) Uncertainties |
| 2. Uncertainties associated with new testing algorithms and test kits | |
| 3. Inadequate training | |
| 4. Work pressures and distractions during times of testing | B) Difficult work conditions |
| 5. Reading test results too quickly or too late | |
| 6. Failure to record and document test results accurately | |
| 7. Stock-outs | |
| 8. Tensions between counsellors and nurses | C) Social interactions |
| 9. Difficult clients | |

## Uncertainties

**Some test-kits cannot be trusted.**   While adequate training reduced uncertainties associated with the test kits, many of our interviewees articulated uncertainties with using some of the test-kits and their associated scripts. Our interviewees preferred some test-kits over others. It was common for interviewees to express which test-kits they preferred or detested. While there was a diversity of views on which tests were 'good' or 'bad', test 1A, the most commonly used test, was repeatedly mentioned as a trustworthy and efficacious test-kit.

> "Test 1B and test 2B would give you a false negative but test 1A continues to be used because we see its efficacy [. . .] Aah test 2B was the worst, it gave out false results." Female nurse, age 45, urban clinic

While test 1A was by far the most favoured test by a majority of respondents, tests 1B, 2B, and 3 were regarded as most problematic, to the extent that some testers felt anxious or even tried to avoid using them. Both tests 1B and 3 were known to be difficult to interpret if the prescribed time window for reading the results was missed. The concern most often mentioned about using test 3 was a lack of training. A HIV counsellor from a rural mission hospital expressed a lack of confidence in using test 3, the tie-breaker on the 2017 version of the national HIV testing algorithm:

> "Test 3 works as a tiebreaker but personally it becomes a challenge for me to use that test. . .' 'kits like test 3 we haven't really fully received training on so that may become a problem because you will end up saying if only I could test someone and they test negative so that I don't go through the struggle of using test 3." Female HIV counsellor, age 40, rural mission hospital

Her lack of training in using test 3, and the fact that she does not use tiebreaker test on a regular basis, may increase the likelihood of her making mistakes when using the test. But it may also amplify her lack of confidence in using the test, which, as she alludes to, could translate into testing bias in order to avoid using test 3 as a tiebreaker. During our observations in one hospital we noted that test 3 was only administered by laboratory staff. Our interviews suggest that tiebreakers were very rare, and we never observed a tiebreaker being performed. While this lack of engagement with test 3 as a tiebreaker suggest non-compliance with the 2017 testing algorithm, the tiebreaker is no longer recommended and has been removed from the testing algorithm. Nonetheless, the message here is that testers' uncertainty with running new or rarer tests (in the context of limited training) can make them avoid using these tests altogether.

One of the difficulties HIV testers face in interpreting results pertains to the readability of lines. As explained by a 42-year-old nurse working on antenatal care in a rural hospital, "there are times when the line on the test kits is faint."From our own observations, we were surprised by how faint the control lines generally were, even on 1A test-kits, which were widely regards as producing 'good' lines. Different explanations were offered to account for faint or unclear lines on the test-kit. One HIV counsellor explained that they once had a batch of 2B kits that produced random and unintelligible lines. Another counsellor explained that a host of different issues may affect how clear the lines will appear:

> "I think it is the test because one could have followed the standard operating procedure on the kit to the dot and still get the faint line. However, at times it is about the client and the anti-bodies in them and then in some cases when one uses too small a blood sample, the

test takes forever to read. So in such cases if one doesn't pay close attention to the time it takes to read the result, one might report a negative result when in actual fact it will be a positive." Female HIV counsellor, age 43, rural hospital

The same counsellor went on to say that besides faint lines, some tests produce multiple lines shortly after the prescribed time window for reading the results:

"I don't know about the quality of some of the kits because soon after 20 minutes multiple lines will start appearing. We end up seeing that there is a kit which is more accurate than other kits." Female HIV counsellor, age 43, rural hospital

Again, some test-kits were highlighted as being particularly prone to producing faint lines, while others, including test 1A, were said always to produce clear lines. This emphasis on the test-kits (rather than the tester) as producing unclear results may, like discussed above, lead HIV testers to note a pattern in certain test-kits producing unclear lines. The accumulation of such experiences and their insecurities with using certain test-kits heightens their distrust, which in turn may affect their inclination to use other test-kits or shape how they interpret test results.

**Uncertainties associated with new testing algorithms.**  Over the past three years, rapid HIV testers in Zimbabwe have endured three changes to the national testing algorithm, and the 2019 WHO guidelines are likely to lead to further iterations. Changes to the national algorithm are spurred by evidence-based recommendations and the pricing and availability of new test-kits. New test-kits are therefore often introduced with changes in the testing algorithm, which contribute to considerable uncertainty and deviation from the script.

"We had an algorithm, which we have stopped using, we now have a new algorithm, but you find that person is following the old algorithm." Male laboratory technician, age 58, district hospital

"The algorithms change every now and then. Even the kits themselves change every now and then. I explained earlier that test 2A was once our tiebreaker but currently 2A is our second test, so they change every now and then. This makes it challenging." Female nurse, age 45, rural health clinic

The occasional change in algorithm meant that testers were not always up-to-date or felt confident in applying the new testing algorithm. For example, at one rural hospital we observed that the same test (1A) was used both for the first screening test and as the confirmatory test when a client tested HIV positive. This practice of solely relying on test 1A was not an isolated incident; it was confirmed in interviews and conversations with testers in other hospitals too. At a District General Hospital, for example, a primary counsellor mentioned that some of their satellite clinics only used 1A test kits because they fully trusted this test to produce the right results. At the same time, we observed that here, in contradiction to the national testing algorithm, that test 2A was used as the first screening test. While deviance from the latest algorithm could sometimes be explained by their confidence in old testing practices and uncertainties associated with new testing practices, occasional stock-outs also led testers to tinker with the algorithm. A primary counsellor from a rural facility, for example, explained:

"We haven't had a second test kit since March. We have test 1A, but we are using it as a second confirmatory test because there is a shortage even at the District level. [. . .] If a client

tests positive, we won't have a second kit. It's been a month... The second test is out of stock." Female counsellor, age 43, rural hospital

The primary counsellor was fully aware of the fact that it would not be possible to re-test the client once the confirmatory test was available again, as they would by then have been initiated on antiretroviral treatment. These pervasive uncertainties and resultant malpractices highlight inadequate levels of training and support.

**Inadequate training.** The preceding findings have already alluded to the importance of training when new test-kits or algorithms are introduced. The lab technician participating in our study stated that "the challenge is to train testers and keep them updated." While all of our respondents had received some level of training in administering HIV test-kits, for some it was more than 10 years ago. New test-kits and algorithms have been introduced since their initial training, and there was agreement amongst our interviewees that refreshers courses were needed. While the Ministry of Health in Zimbabwe continually offers refresher courses, they were described as becoming increasingly rare due to funding constraints. When refreshers courses were organised, it was often for a selected few who, in return, would be expected to pass on their learning to other testers in their health facility. A number of our interviewees drew a direct link between inadequate training and risks of misdiagnosis, as exemplified by one HIV counsellor:

"If you did not go for training then it is difficult to administer the rapid tests. I remember at one point there was a certain individual that did not go for training and they wanted to test someone, and he did it incorrectly. If one goes for training it is very easy because you will be taught the correct procedure of administering these tests and all that is involved to obtain the correct results." Female HIV counsellor, age 42, rural mission hospital

It is clear from our findings that pervasive uncertainty from HIV testers' trust in the test-kits, frequent changes in testing algorithm and test-kits, and limited training in using new test-kits and algorithms, introduces biases and a tinkering of recommended testing practices, which are potential sources of misdiagnosis.

## Difficult work conditions

A nation-wide reduction in the cadre of healthcare workers, coupled with successful HIV testing campaigns, puts pressure on HIV testers. A nurse encapsulates this pressure succinctly:

"At times we improvise; we work with what is there. We are forced to work under harsh conditions. At times we don't execute our duties as we are supposed to due to shortage of staff and resources." Female nurse, age 40, urban clinic

All interviewees spoke about these difficult work conditions and the potential implications of resource-constrained conditions for misdiagnoses.

**Work pressures and distractions during times of testing.** Our interviews repeatedly mentioned the high volumes of people presenting at their health facilities for HIV testing. We also noted that much time was spent on paperwork and report writing, especially for non-governmental organisations (NGOs) that had funded testing activities. The workload fluctuated, with early and late weekdays being particularly busy. Some testers performed up to 50 HIV tests on a busy morning. Given a theoretical maximum of three tests per tester per hour, as each test typically takes 15–20 minutes to read and must be accompanied by pre-and post-test counselling, this is illustrative of a heavy workload–also expressed through accounts of long

queues, which put pressure on HIV testers to hurry up, skip lunch, and develop strategies for testing multiple clients at once. In Gutu district for instance, we observed how counsellors routinely tested several people at once, lining the tests up on the table, thus increasing the risk of mixing up results. As a HIV counsellor explains, the burgeoning workload may result in less than perfect HIV testing:

> "Yes the workload may become too much since there will be many patients and you do not have time to do everything perfectly. Imagine, the patients will be standing outside in a long queue, you have limited time and the only tester, and the different test kits revealing a different set of results." Male HIV counsellor, age 50, rural mission hospital

This counsellor also alludes to a dilemma mentioned by a number of participants, namely, the challenge of dealing with the follow-up required from a first positive test result. Subsequent tests would need to be carried out to confirm or disconfirm the first test result, and if positive, the client would require counselling and referral–all of which would cause further delays for the people waiting. This may contribute to a potential bias against testing someone HIV positive. There were also numerous accounts and observations of how HIV testers were distracted or called for urgent matters, disrupting and confusing the testing process. This ranged from nurses, who in the middle of testing and counselling, were called to attend emergency cases, to counsellors being interrupted by phone calls, text messages, or people bursting into the testing room with questions or handing them files to check.

**Reading test results too quickly or too late.** Difficult work conditions, and the heavy workload in the particular, may, as explained by one nurse, "cause you to read the results before the actual time has lapsed." A male HIV counsellor echoed this, linking 'acting quickly' with work pressures and failure to follow SOPs:

> "A person may be quick, perhaps because of work pressures. I'm just saying what can happen if a person does not follow procedure. They may draw too much blood, when only a certain amount of blood is needed. They may dilute the blood more than necessary or add too little buffer. They may read the results when the time hasn't lapsed or read them much later, long after the 15 minutes you have to wait." Male HIV counsellor, age 33, rural hospital

One consequence of reading test results too quickly or too late is misdiagnosis, as explained by a lab technician: "I have seen others read results after a short time, before the line had reached the control line. They read it within some minutes or seconds. What happens then is misdiagnosis." We made similar observations, noting that in a number of health facilities, testers found it difficult, due to either workloads or disruptions, to read the results in the prescribed time window. Although our interviewees stressed the importance of reading test results after 15 minutes, our observations noted quite lax attitudes to time keeping, even in high-performing facilities. We did not see HIV testers set an alarm to make sure they would keep track of all the different tests they were running at the same time, and sometimes observed testers reading and recording results immediately after administering the test.

**Failure to record and document test results accurately.** While many HIV testers felt confident with the procedures for coding and linking clients with test strips and record books and claimed that misclassifications were rare, they acceded it could happen, and "if it happens then it will probably be due to a lapse of concentration or something like that" (Male nurse, age 36, district hospital). Failure to record and document test results accurately was often attributed to 'being tired' or the pressures the testers work under. One HIV counsellor explains

how misclassification mistakes are more likely to occur when they line up test-kits on a table and bring in groups of people for testing:

> "If you are administering tests to a number of people [. . .] there will be a lot of test strips. If you do not label the test kits [. . .] you might get confused when issuing the results." Female HIV counsellor, age 43, rural hospital

We observed that often only the client's initials were recorded on the test scripts, which carries a high risk of mixing up results when testing multiple people at once. We also observed occasions when blood was spilled on the table, heightening the potential for contamination of test strips.

A nurse attributed the actual workload associated with the registers as a potential source of misclassification.

> "'How easy is it to record results. . .?' 'As an individual it's not easy [. . .] we will be four, with one person testing, and the other recording in the HTS book and another one recording in the Syphilis book and the other one recording in the patients' cards. [. . .] The registers are too many for a nurse [. . .] you end up having a burnout. It's very difficult like if a person tests positive aaah then you're in big trouble [. . .] then aaaah there's a lot of work." Female nurse, age 45, urban clinic

This sharing of the recording work between a team of several people was confirmed by our observations.

**Stock-outs.**   In line with a recent HIV service evaluation in Zimbabwe, which found most health facilities to report at least one test-kit stock-out in the prior year [15], most of our participants spoke about instances of test-kit stock-outs. Earlier we presented the stock-out experiences of a 43-year-old HIV counsellor. Reflecting on supply chain deficiencies, she explained how they had run out of second confirmatory tests for an extended period of time, and how the shortage was district-wide, making it impossible for them to borrow test-kits from neighbouring health facilities. In response to the stock-outs for confirmatory tests, she went on to explain that "the sister in charge. . . gave us the go ahead to administer the tests [outside of the algorithm] and said that it is better for the client to be initiated on ART." By using test 1A as both a first and confirmatory test, the stock-out forced them to deviate from the HIV testing algorithm, which may lead to misdiagnoses. The nurse recognises "there is no quality in it, but we don't have any option.". However, the approach of another hospital to deal with stock-outs was not to tinker with the HIV testing strategy, but simply to send clients home, fully aware of what this might mean for the clients:

> "The problem I mentioned earlier on, I think it is important for kits to always be readily available so that there are no times when a client gets turned back due to lack of kits. When people get turned back they go back to their homes and get sick and are vulnerable to opportunistic infections like TB. Some cases the CD4 count would lower so much and the person dies whereas this would not have happened had the kits been available." Female nurse, age 42, rural hospital

During our observations we noticed this resource-depletion in other ways too, with gloves that kept tearing, the lack of air-conditioning in drug storage rooms, and a general chaotic storage of medicines and equipment.

## Social interactions

**Tensions between counsellors and nurses.** In Zimbabwe, the relationship between primary counsellors and nurses is complex. Whilst we observed numerous working relationships, we also observed some tensions between them, with nurses occasionally refusing to help counsellors with HIV testing even if the demand was high. The tensions arise from pay differentials and recent task shifting. To meet the demand for HIV testing in Zimbabwe, and with the introduction of point-of-care HIV testing, the task of HIV testing shifted to counsellors, moving them closer to one aspect of the clinical work of nurses. This change in role and responsibility was highly motivating for our participating counsellors, and often accompanied with lucrative employment contracts with NGOs. HIV testers, even within one health facility, were therefore paid very different salaries for the same work, depending on who their employer was. For many registered and state-employed nurses, who are paid less than NGO-employed counsellors despite their higher educational qualifications, this was a source of much discontent and jealousy, affecting their working relationship.

**Difficult clients.** We invited our interviewees to reflect on challenges they encountered during the testing process, and sometimes this led to a discussion on clients with "anger issues" who were "intimidating". Men in particular were described as intimidating, resulting in a fear of issuing a positive test result to certain groups of men.

> "What I like the least is when I administer a test to someone and then he tests positive. As soon as I tell him that his result is positive, he just stands up, opens the door and leaves the room. He goes back home and I won't know if he has any suicidal thoughts or if he is blaming someone for it. So, you will be in suspense [. . .] He won't be violent as such but you will see that he is angry." Female HIV counsellor, age 36, mission hospital

Men were also described as clients who wanted the HIV testing to be quickly done and over with, rushing the testers. Although none of our interviewees conceded that the fear of issuing a positive testing result to men, or being rushed by men, affected the quality and authenticity of their HIV testing, it is nonetheless likely to introduce testing biases. HIV testers may for instance be inclined to side with a negative test result if the result is ambiguous.

Similar biases may be introduced during couples testing. A significant number of our interviewees spoke about the challenges of issuing discordant test results to couples, being fully aware that this may lead to break-ups or worse.

> "If we test a couple and one person tests positive and the other one tests negative especially in favour of the husband you will observe that the relationship has been affected. The partner will be in fear of getting dumped just because they have tested positive, yet their husband hasn't tested positive so those are some of the challenges that are happening to us, that some marriages are breaking down." Female HIV counsellor, age 40, mission hospital

A more generic group of "difficult clients" included anyone who was either not ready to get tested (e.g., expectant mothers, for whom testing is part of antenatal care) or ready to accept their sero-conversion. Men, discordant couples and 'difficult clients' required more time and counselling, exacerbating the workload. While none of our interviewees spoke about feeling pressured to interpret results incorrectly, our findings did note social interactions and circumstances that are highly challenging for HIV testers, and which may contribute to testing biases, particularly in busy and resource-depleted health facilities.

## Discussion

Point-of-care rapid diagnostic HIV tests have made it possible for healthcare workers at any level to test, avail results, and initiate timely treatment, if necessary, thus supporting global testing and treatment targets. However, as outlined in the introduction, disconcerting levels of misdiagnosis have been observed in sub-Saharan Africa, including Zimbabwe [23]. We set out to investigate potential pathways to misdiagnosis, exploring the accounts and practices of rapid HIV testers in Zimbabwe. The accounts constituted rapid HIV test-kit and algorithm uncertainties, difficult work conditions as well as certain social interactions as three particular kinds of 'problems', which may contribute to deviances from the official testing procedures, as well as testing and interpretation biases. We discuss each of the 'problems' in turn.

Our interviewees preferred some test-kits over others, expressing a lack of confidence in using certain test-kits. There appeared to be a correlation between their confidence in test-kits and their frequency of use. Test 1A, a first screening test, was generally favoured, whilst other less frequently used tests were regarded more challenging. This is problematic given their reliance on other rapid HIV tests to determine the authenticity of positive test results. Inadequate training on new or less frequently used rapid HIV tests and regular changes in national HIV testing algorithms further heightened these uncertainties. The uncertainty warranted a tinkering with national HIV testing algorithms, for instance choosing confirmatory tests they trusted or were familiar with, or interpretation biases by hoping for negative or concordant results in order to avoid infrequently used tie-breaker tests. Whilst our study supports previous research highlighting that a lack of continuous, ongoing proficiency training can diminish the authenticity of point-of-care test results [24, 25], we provide additional insight by unpacking perceptions of causes and consequences of the uncertainties that arise from inexperience and changes in the script.

We found a combination of high workloads, staff shortages, and supply chain deficiencies to affect rapid HIV testing practices. This is in agreement with the broader literature on point-of-care HIV testing [26], including a review of 18 qualitative research studies exploring the views and experiences of healthcare workers from a broad range of countries and settings [11]. Specifically, we found difficult work conditions to compound the uncertainties discussed above, and to be exacerbated by the growing number of people seeking testing. It was difficult for HIV testers to follow the official testing procedures, feeling the pressure of people waiting to get tested, or getting distracted by other tasks. We observed testers taking shortcuts, testing multiple people at once, and having fairly lax attitudes regarding the time keeping of tests. The pressures felt by testers may contribute to a potential bias against testing someone HIV positive, as this would create further work. The difficult work conditions also affect the social interactions that take place at the health facilities. Tensions between registered nurses and primary counsellors, manifested in power and pay differentials, were occasionally (not always) observed to obstruct good working relationships around point-of-care testing. In South Africa, similar tensions have been observed following the task shifting of testing practices from nurses to more lay health workers [27]. We also noted that testers feared certain client groups, who required significant counselling time, or were either intimidating or emotionally challenging, such as couples with discordant test results. Other studies have highlighted the emotional burden of HIV testing on healthcare workers [26, 28].

These three contexts, in different ways, highlight frictions that can prevent HIV testers from adhering to the national testing strategy and SOPs (the script). HIV testers might side with a negative test result, for instance if the result is ambiguous, or read the test strip before time, in order to save time, reduce the work load, avoid confrontation, or to be the bearer of bad news that may result in a marriage break-up. By approaching (dis)engagement with the

national testing algorithm as a social practice [29], we have unpacked a range of factors and bundles of practices whose configuration steer and govern that capacity of HIV testers to follow their script.

It is widely believed that point-of-care tests should be simple and meet the ASSURED criteria: affordable, sensitive, specific, user friendly, rapid and robust, equipment-free and delivered [24, 30]. Our study has availed numerous insights that have implications for the policy and practice of a number of these criteria, but also identified a number of structural criteria that affect the success of point-of-care technologies. Our interviewees did not find all rapid HIV tests user friendly and easy to use with minimal training. Equally, not all tests provided clear and actionable results, providing the HIV testers with room for interpretation. More and better training, as well as regular refreshers courses for all HIV testers, with less reliance on 'cascading' training within health facilities needs to be factored into point-of-care testing programmes. Future training, as well as SOPs, should be extended to provide more guidance on what to do when test results are ambiguous, in the event of stock-outs, or how to test different categories of clients. Our findings also suggest that guidance on how to test multiple people at once may be necessary, and that HIV testers, in the context of the 'problems' we have identified, are given time to reflect on their own testing and interpretation biases.

External quality assessment (EQA), as recommended by the WHO [9], was being rolled out in Zimbabwe in 2017/18 and reported as being present in 25% of cases at the time of our surveillance [23]. Although EQA coverage might have been higher at the time of the qualitative data collection, it did not feature in our findings. Echoing concerns raised by Stevens et al. [31] our findings reveal that existing point-of-care HIV testing protocols and quality controls in Zimbabwe remain inadequate considering both the volume of HIV tests and the reality of resource-depleted health facilities. There is a need for innovative and scalable quality controls that are both appropriate and go beyond laboratory quality controls to meet the complex realities point-of-care testers encounter. In other words, there is a need to re-think the kind of standards and checklists that can realistically be introduced in real-life settings. There is also need for further training of HIV testers, accompanied with a certification system. The Zimbabwe Ministry of Health and Child Care and partners are currently strengthening the national testing algorithm, proficiency testing and their EQA systems, all of which are relevant to addressing the problems identified in this paper. However, quality control challenges are not limited to sub-Saharan African countries. A study looking at the feasibility of using rapid HIV tests in general practice in France identified impediments similar to those that we have observed: misinterpretation of test results, complexity of quality control, and lack of training [32]. It is critical that 'improvements' being considered by WHO and governments, are run past the cadre of healthcare workers who perform rapid diagnostic HIV testing on a daily basis. They need to be consulted, both to identify 'improvements' that resonate with their everyday realities, but also to understand how broader evidence-informed 'improvements' might be received by the testers.

Finally, our findings speak to new developments in the donor community, where a shift is taking place away from broad population wide targets (e.g., the percentage of people tested) towards an emphasis of 'yields' of HIV positive results and greater targeting. This change in emphasis could potentially lead to less testing overall (reducing workloads in the clinic) but more work in identifying and reaching the targets (increasing workloads for community health workers). While donors may see this as a cost-saving exercise, it may alleviate or flat-line the workload experienced by HIV testers and improve the quality of their work. Future research would need to explore what the net effect of such a strategic shift might be on workloads, diagnostic errors, and treatment targets.

Our findings are constrained by some methodological limitations, which deserve mentioning. First, the study relies primarily on self-reported data. Although we sought to observe what

happens in clinical practice, our rapid ethnographic approach may have undermined our wish to go beyond the observation of more performative practices. Future research in this area of study may consider adopting a more in-depth and longitudinal ethnographic approach. Second, and relatedly, our participants could have wanted to protect their own, or their organisation's reputation, meaning the study was susceptible to social desirability bias, with participants representing themselves in particularly positive way. Third, our study was cross-sectional and only provides a brief snapshot into HIV testers experiences and practices of using rapid HIV tests at a particular moment in time. The timing of data collection coincided with recent changes in the national algorithm and associated test kits in Zimbabwe. Longitudinal research, unpacking the dynamic and changing nature of testers' experiences and practices would be useful. Fourth, the generalizability of our findings is limited and may not apply to other settings, as the structure and delivery of HIV testing services will vary from others. This study only explored the perspectives of HIV testers. Future research could usefully broaden its scope and include the perspectives and experiences of policy makers as well as health facility administrators.

## Conclusion

While there are numerous studies exploring the experiences and perspectives of healthcare providers involved in point-of-care HIV testing [11], this study, as one of the first, has drawn on such perspectives to unpack pathways to misdiagnosis in Zimbabwe. We have found that uncertainties, difficult work conditions, and certain social interactions contribute to testing and interpretation biases, as well as deviances from the official testing procedures, which may offer some explanation to high levels of misdiagnosis. There is a need to take heed of the fact that our interviewees located deviance from HIV testing procedures in this broader context, and to caution against user error explanations to misdiagnosis in the absence of a broader recognition of structural determinants.

These findings provide important learning opportunities, not only for point-of-care HIV testing programmes, but also for the roll-out of rapid point-of-care testing for emerging infectious diseases, such as SARS-CoV-2. Our findings demonstrate that mere availability of lab space and test kits is insufficient in providing good quality, reliable test results, which are vital in managing the spread of infection. Considering that very large numbers of SARS-CoV-2 tests–potentially using several different new rapid diagnostic test kits, SOPs, and testing algorithms–will be needed in resource-limited settings which could be overwhelmed if a serious COVID-19 outbreak occurs, it will be extremely important to take into account the clinical care settings where these tests are administered, and to adjust guidance, training and resources according to local needs and in collaboration with the healthcare professionals who will perform these tests.

## Acknowledgments

We would like to thank all the participants who contributed their time and effort to the study. We would also like to thank Melinda Moyo, Rangararirai Nyamwanza, and Constance Makumbe for their fieldwork and research assistance. The content is solely the responsibility of the authors and does not necessarily represent the official views of the Bill and Melinda Gates Foundation.

## Author Contributions

**Conceptualization:** Morten Skovdal, Nadine Beckmann, Constance Nyamukapa, Simon Gregson.

**Data curation:** Nadine Beckmann, Rufurwokuda Maswera, Constance Nyamukapa.

**Formal analysis:** Morten Skovdal, Nadine Beckmann.

**Funding acquisition:** Constance Nyamukapa, Simon Gregson.

**Investigation:** Constance Nyamukapa, Simon Gregson.

**Methodology:** Constance Nyamukapa, Simon Gregson.

**Project administration:** Rufurwokuda Maswera, Simon Gregson.

**Resources:** Simon Gregson.

**Supervision:** Morten Skovdal, Nadine Beckmann, Rufurwokuda Maswera, Constance Nyamukapa, Simon Gregson.

**Writing – original draft:** Morten Skovdal.

**Writing – review & editing:** Morten Skovdal, Nadine Beckmann, Rufurwokuda Maswera, Constance Nyamukapa, Simon Gregson.

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
