## [Decision Letter · Decision Letter 0]

22 Jun 2020

PONE-D-20-02895

Uncertainties, work conditions and testing biases: Potential pathways to misdiagnosis in point-of-care rapid HIV testing in Zimbabwe

PLOS ONE

Dear Dr. Skovdal,

Thank you for submitting your manuscript to PLOS ONE. After careful consideration, we feel that it has merit but does not fully meet PLOS ONE’s publication criteria as it currently stands. Therefore, we invite you to submit a revised version of the manuscript that addresses the points raised during the review process.

Please address all of the concerns of the reviewer. 

We look forward to receiving your revised manuscript.

Kind regards,

Julie AE Nelson, PhD

Academic Editor

PLOS ONE

Journal Requirements:

2. Please include additional information regarding the interview guide used in the study and ensure that you have provided sufficient details that others could replicate the analyses. For instance, if you developed a guide as part of this study and it is not under a copyright more restrictive than CC-BY, please include a copy, in both the original language and English, as Supporting Information. In addition, please provide details of any pilot testing of the guide that took place.

4. Please include your tables as part of your main manuscript and remove the individual files. Please note that supplementary tables (should remain/ be uploaded) as separate "supporting information" files.

Reviewers' comments:

Reviewer's Responses to Questions

**Comments to the Author**

1. Is the manuscript technically sound, and do the data support the conclusions?

Reviewer #1: Yes

2. Has the statistical analysis been performed appropriately and rigorously? 

Reviewer #1: N/A

3. Have the authors made all data underlying the findings in their manuscript fully available?

Reviewer #1: No

4. Is the manuscript presented in an intelligible fashion and written in standard English?

Reviewer #1: Yes

5. Review Comments to the Author

Reviewer #1: With the availability of point – of – care diagnostic technology, globally, HIV testing coverage has been significantly improved through task shifting and decentralization of the service. While this facilitate access to HIV testing, greater attention to quality assurance is required and quality management system need to be in placed to ensure the quality of HIV testing.

This qualitative study provided very important and practical perspectives on HIV misdiagnosis in Zimbabwe. The potential factors that may lead to misdiagnosis were classified in three thematic topics: uncertainties, work condition and testing bias. Such information would be very useful for Zimbabwe (and may be other country that have similar context) in terms of policy development or actions to be taken to ensure HIV testing quality. Overall, the manuscript was very well written. I have several minor comments for the authors to consider:

- Page 7, age of the researchers is still missing “x to x”. However, I don’t think it is necessary to report researchers’ age.

- Figure 1. Need to have descriptive caption. In the text, it said Figure 1 I national testing algorithm. However, from what I see in the Figure, this is national testing strategy not testing algorithm.

- Table caption should be placed above the tables

- Study participants: The Table 1 indicated 18 hospitals involved in the study. While the number of HIV counsellors and nurses interviewed were 17 and 10 respectively, only one lab technician participated in the study. The author said two or three rapid HIV testers were recruited from each facility (from 11 health facilities) who met the inclusion criteria. However, it is unclear if each facility had more than two (or three) testers who were eligible for participation in the study, how the interviewees were selected. The authors may need to provide further details.

- External quality assessment (EQA) is a critical component for quality assurance system. It would be good if the authors can discuss whether EQA was available in Zimbabwe or not. The authors reported testers received some training but whether there was a certification system (for the testers and for the testing sites) in the country?

- Reference: WHO handbook on “Improving the quality of HIV -related point-of-care testing: ensuring the reliability and accuracy of test results” could be a relevant reference for this manuscript.

6. PLOS authors have the option to publish the peer review history of their article (what does this mean?). If published, this will include your full peer review and any attached files.

Reviewer #1: Yes: Dr Van Thi Thuy nguyen

---

## [Author Response · Author response to Decision Letter 0]

1 Jul 2020

- Page 7, age of the researchers is still missing “x to x”. However, I don’t think it is necessary to report researchers’ age. 

That was an error, thank you. We agree that age is not relevant and have removed the reference to their age. 

- Figure 1. Need to have descriptive caption. In the text, it said Figure 1 I national testing algorithm. However, from what I see in the Figure, this is national testing strategy not testing algorithm. 

We have added a descriptive caption to Figure 1. 

We have also amended the text to specify that it is a strategy. Because we anonymized the tests, the reviewer helpfully pointed out that it is no longer a populated algorithm, but a strategy. We also now explain this to the reader. 

- Table caption should be placed above the tables 

Done

- Study participants: The Table 1 indicated 18 hospitals involved in the study. While the number of HIV counsellors and nurses interviewed were 17 and 10 respectively, only one lab technician participated in the study. The author said two or three rapid HIV testers were recruited from each facility (from 11 health facilities) who met the inclusion criteria. However, it is unclear if each facility had more than two (or three) testers who were eligible for participation in the study, how the interviewees were selected. The authors may need to provide further details. 

Thank you for this comment. We have added a sentence in the methodology to clarify this point:

“As more than two or three HIV testers were eligible for participation at many of the health facilities, we invited the local Matron or Sister in charge to select the participants based on their experience and availability to participate in an interview on the day of our visit.”

- External quality assessment (EQA) is a critical component for quality assurance system. It would be good if the authors can discuss whether EQA was available in Zimbabwe or not. The authors reported testers received some training but whether there was a certification system (for the testers and for the testing sites) in the country? 

In the discussion we now explain:

“External quality assessment (EQA), as recommended by the WHO [9], was being rolled out in Zimbabwe in 2017/18 and reported as being present in 25% of cases at the time of our surveillance [20]. Although EQA coverage might have been higher at the time of the qualitative data collection, it did not feature in our findings.”

We have also added a note on the absence of a certification system to the discussion:

“There is also need for further training of HIV testers, accompanied with a certification system (this is currently under consideration in Zimbabwe).” 

- Reference: WHO handbook on “Improving the quality of HIV -related point-of-care testing: ensuring the reliability and accuracy of test results” could be a relevant reference for this manuscript. 

We have incorporated the handbook into our introduction (in relation to WHO recommendations) and into the discussion (to talk about EQA).

---

## [Decision Letter · Decision Letter 1]

23 Jul 2020

Uncertainties, work conditions and testing biases: Potential pathways to misdiagnosis in point-of-care rapid HIV testing in Zimbabwe

PONE-D-20-02895R1

Dear Dr. Skovdal,

We’re pleased to inform you that your manuscript has been judged scientifically suitable for publication and will be formally accepted for publication once it meets all outstanding technical requirements.

Kind regards,

Julie AE Nelson, PhD

Academic Editor

PLOS ONE

Additional Editor Comments (optional):

There are grammatical errors that must be corrected before publication. See the list below.

Throughout, “test-kits” does not need to be hyphenated, but if it is, it must be consistent, including in Table 2.

Line 32: add comma after “facilities”

Line 66: change to “the recent”

Line 70: move comma to before “as”

Line 89: Add “Zimbabwe” as follows: “to the current Zimbabwe national”

Figure 1: also add “Zimbabwe” as follows: “The Zimbabwe National HIV”

Line 92: delete “illustrated in figure 1” as redundant to previous sentence

Line 110: remove “either” and add comma after “engineer”

Line 112: change “HIV tests and the standard operating procedures and testing algorithms” to “HIV tests, SOPs, and testing algorithms”

Line 115: fix misspelling of “assess”

Line 119: add comma after “materials”

Lines 123-126: change all verbs to past tense

Lines 134-145: move this paragraph to the introduction; make the last part of the introduction about why you have done this study and include the information in this paragraph.

Line 147: Change first word to “This”

Lines 147-152: change all verbs to past tense

Line 183: add comma after “test-kits”

Line 195: change “as well as” to “and”

Line 197: add comma after “pharmacy”

Line 224: change “is” to “was”

Line 229: change “standard operating procedures” to “SOPs”

Line 243: add comma after “2B”

Line 247: remove comma after “hospital”

Line 277: Add period after “positive”

Line 304: change dashes to parentheses

Line 331: remove comma after “few”

Line 340: change “, whether it relates to” to “from”

Line 342: add “s” to “introduce”

Line 382: change “standard operating procedures” to “SOPs”

Line 388: add period after “wait”

Line 409: add period after “results”

Lines 432-435: this is the same quote as used in Lines 314-317. Do not use this quote twice; either use a different quote in one of the locations or refer to the earlier quote here without repeating it.

Line 534: change “standard operating procedures” to “SOPs”

Line 536: add comma after “confrontation”

Line 549: change “standard operating procedures” to “SOPs”

Line 556: change “et al” to “et al.”

Reviewers' comments:

Reviewer's Responses to Questions

**Comments to the Author**

1. If the authors have adequately addressed your comments raised in a previous round of review and you feel that this manuscript is now acceptable for publication, you may indicate that here to bypass the “Comments to the Author” section, enter your conflict of interest statement in the “Confidential to Editor” section, and submit your "Accept" recommendation.

Reviewer #1: All comments have been addressed

2. Is the manuscript technically sound, and do the data support the conclusions?

Reviewer #1: Yes

3. Has the statistical analysis been performed appropriately and rigorously? 

Reviewer #1: N/A

4. Have the authors made all data underlying the findings in their manuscript fully available?

Reviewer #1: No

5. Is the manuscript presented in an intelligible fashion and written in standard English?

Reviewer #1: Yes

6. Review Comments to the Author

Reviewer #1: The authors have sufficiently addressed all the recommendations. It can be considered for publication.

7. PLOS authors have the option to publish the peer review history of their article (what does this mean?). If published, this will include your full peer review and any attached files.

Reviewer #1: **Yes: **Van Thi Thuy Nguyen

---

## [Editor Report · Acceptance letter]

4 Aug 2020

PONE-D-20-02895R1 

Uncertainties, work conditions and testing biases: Potential pathways to misdiagnosis in point-of-care rapid HIV testing in Zimbabwe 

Dear Dr. Skovdal:

I'm pleased to inform you that your manuscript has been deemed suitable for publication in PLOS ONE. Congratulations! Your manuscript is now with our production department. 

Kind regards, 

on behalf of

Dr. Julie AE Nelson 

Academic Editor

PLOS ONE